# Infrared Laser Application to Wood Cutting

**DOI:** 10.3390/ma13225222

**Published:** 2020-11-19

**Authors:** Monika Aniszewska, Adam Maciak, Witold Zychowicz, Włodzimierz Zowczak, Thorsten Mühlke, Bjoern Christoph, Samir Lamrini, Sławomir Sujecki

**Affiliations:** 1Department of Biosystems Engineering, Warsaw University of Life Sciences-SGGW, Nowoursynowska 164, 02-787 Warsaw, Poland; monika_aniszewska@sggw.edu.pl (M.A.); witold_zychowicz@sggw.edu.pl (W.Z.); 2Laser Processing Research Centre, Kielce University of Technology, al. Tysiąclecia Państwa Polskiego 7, 25-314 Kielce, Poland; wzowczak@tu.kielce.pl; 3LISA Laser Products GmbH, Albert-Einstein-Straße 4, 37191 Katlenburg-Lindau, Germany; tmuehlke@lisalaser.de (T.M.); bchristoph@lisalaser.de (B.C.); slamrini@lisalaser.de (S.L.); 4Department of Telecommunications and Teleinformatics, Faculty of Electronics, Wroclaw University of Science and Technology, Wybrzeże Wyspiańskiego 27, 50-370 Wroclaw, Poland; slawomir.sujecki@pwr.edu.pl

**Keywords:** laser, wood, laser shears, laser wood cutting

## Abstract

While lasers are widely used across various industries, including woodworking, few studies to date have addressed the issue of cutting fresh wood. In the present investigation, wood stemming from fresh tree branches was cut at different laser powers and beam travel speeds. A fiber laser and a CO_2_ laser were used for the research. The cellular structures of the cut surfaces were examined, with some of them found to be covered with a layer of compacted, charred cells. This may be a favorable phenomenon, preventing the invasion of pathogens via the wounds caused by laser beam branch cutting in nurseries, plantations, and orchards.

## 1. Introduction

Over the years, a variety of laser types have been developed at research centers around the world, including gas, rod, fiber, disk, and semiconductor lasers [1,2,3]. Laser technology is currently employed in the automotive, shipbuilding, aviation, armaments, and energy industries, as well as in medicine, for the purposes of welding, hardfacing, melting, surface heat treatment, soldering, penetrating, engraving, ablating, and more. Thus, laser light has many applications in the processing of various materials [4]. In particular, laser light technology, next to water jet and ultrasound, offers an alternative method for wood processing [5].

The cutting of wooden elements was one of the first commercial applications of lasers in the early 1970s [6,7,8]. Laser light is now used in woodworking for marking, engraving, surface cleaning [9], primary wood processing in automated systems [10], and MDF (medium-density fiberboard) cutting [11,12]. Lasers are widely applied in the furniture industry (e.g., in the production of inlay elements) and toy industry and in the manufacture of decorations and ornaments. The advantages of laser technology include fast processing of curvilinear elements and high durability of the tool itself, as the laser beam does not melt or become dull, and so it does not require regeneration. Lasers are safe to operate because they do not have fast-rotating elements, which often lead to accidents. Furthermore, laser processing is chip-free and does not produce shavings or dust. However, despite the advanced technology, lasers also have certain drawbacks, which limit their use in the woodworking industry. Laser light is absorbed by the two main ingredients of wood, for instance, cellulose and lignin in the case of a CO_2_ laser, with the result of converting laser energy into heat. Given the low thermal conductivity of wood, this damages wood tissue, with hot gasses giving rise to charcoal deposits on the surface of the processed material [13,14,15]. The thickness of the charred layer depends on the physical properties and size of the workpiece, processing orientation (along or across the grain), and laser parameters (mostly wavelength and power density). According to [16], the thickness of the charred layer is proportional to wood density. In the study of Swaczyna and Beer [14], laser cuts made across the grain were charred to a greater extent than those made along the grain, which is attributable to wood structure. The shape of the laser cut is associated with the focal point of the beam. As reported by Tayal et al. [17], the optimum location of the focal point is below the surface of the workpiece, as then the sides of the cut are more regular and less charred as compared with focusing laser beams, for example, on the surface of the workpiece.

Analysis of the transmittance spectrum of wood shows the wavelengths that are most effective for laser wood processing [18]. The devices most widely used in woodworking are molecular CO_2_ lasers emitting radiation with a wavelength of 10.6 μm (wave number 943 cm^−1^), which is well absorbed by cellulose [5].

Efforts are underway to apply Er:YAG solid-state lasers emitting radiation with a wavelength of 2.94 μm (wave number 3401 cm^−1^), which is close to the next favorable absorption bandwidth for wood, which is, 2.937 μm (3405 cm^−1^) [19]. Another wavelength of potential interest is at around 3000 nm (Figure 1). There is intensive research being conducted at the moment aiming to develop coherent light sources within this wavelength range with the maximum output power and maximum pulse energy increasing each year [20,21,22]. Hopefully, lasers operating near 3000 nm will soon become widely used for wood cutting, trimming, and processing.

A major advantage of Er:YAG lasers is that their radiation is absorbed by water [23].

The use of lasers with even shorter wavelengths has been suggested, and in particular 355 nm (wave number 28,169 cm^−1^) [24]. While the laser cutting of dry wood has been described by numerous authors [25], to date no reports have been published on the use of lasers for cutting fresh shoots or branches in orchards, plantations, or nurseries, e.g., to prune trees and shrubs. Semiconductor technology provides opportunities to develop mobile laser devices for wood cutting that would be characterized by sufficient beam power and acceptable dimensions. Such lasers may not produce very high quality beams, but that parameter is not very important in the case of cutting shoots and branches; furthermore that drawback would be more than compensated by their compact design and relatively low price. 

Tree trimming and pruning is a common practice both in nurseries of ornamental trees and in orchards to ensure better growth and fruit yield. Currently, the tools used for these purposes include pruning shears and saws, which damage the wood structure and cellular integrity. Therefore, cutting with those implements must be followed by spraying the affected plants with an antimicrobial agent to prevent an infection [26]. On the other hand, laser application may prevent excessive damage to the wood structure while charring the cut surface, decreasing the risk of pathogens invading the wound site, which may reduce the need for chemicals. It appears that a fiber-optic laser would be optimal [27] since the work head could be manipulated away from the beam source. Therefore, preliminary trials involved this type of laser [28] with a pulsed and continuous-wave (CW) beam used for penetrating pine (*Pinus sylvestris* L.), birch (*Betula pendula* L.), beech (*Fagus sylvatica* L.), and oak (*Quercus robur* L.) roundwood samples with bark along and across the grain. Two pulsed laser beams with a diameter of 4 mm and wavelength of 800 nm (with an energy of 4.5 and 2.0 J and a frequency of 22 and 30 Hz) were applied for penetrating the pine wood.

Finally, for the sake of completeness, one should also mention the sound generation occurring during laser ablation, which can be used effectively for designing an effective control system aiding material processing with pulsed lasers [29].

The objective of the study was to determine the effectiveness of the laser beam in processing wood stemming from fresh tree branches while using different parameters of the laser beam and different beam travel speeds.

## 2. Methodology

The study involved a comparison of cut surfaces made with laser beams set to different power levels and travel speeds with kerfs made with garden shears and saws (implements that are typically used for cutting shoots and branches in orchards).

Our initial study involved a 150 W CW thulium-doped silica glass laser operating near 2000 nm with and without a converging lens and a 210 W CW laser with a converging lens. The less powerful laser was used to penetrate pine wood samples across the grain, while the more powerful one was additionally applied to beech, birch, and oak wood samples along and across the grain. The laser cavity is formed with a fiber Bragg grating (FBG) at one end of the fiber and a perpendicularly cleaved fiber end at the other one. The perpendicularly cleaved fiber end provides feedback via the Fresnel reflection phenomenon.

The laser beam was set to penetrate the middle of the stem (but outside the pith area) for the longitudinal orientation (Figure 1a) and the largest diameter for the transverse orientation (Figure 1b).

The experimental station consisted of a semiconductor pumping laser with a wavelength of 788 nm (1), a fiber laser resonator consisting of a fiber Bragg grating (2), a fiber splice connection (3), a Tm-doped fiber (4), a collimator (5), a filter (HR 788 nm/AR 2 μm) (6), and a converging lens (7) (Figure 2). The optical system produced a focused output beam with a wavelength of 2 μm.

The results of the measurements of the impact of the fiber laser beam on wood are presented in Table 1. The 150 W CW laser beam without a converging lens was not able to penetrate 120 mm diameter pine samples across the grain. The same beam applied in conjunction with a converging lens with a focal length of 500 mm penetrated the sample over 129 s when focused on the sample surface and over 243 s when the focal point was located in the middle of the sample.

The 210 W CW laser beam with a focal length of 230 mm penetrated pine wood samples across the grain over 3 s in the first trial and 5 s in the second trial. Furthermore, it penetrated 120 mm pine samples along the grain over 1 s in two trials.

The same laser beam penetrated 120 mm diameter beech wood samples across the grain over 11, 4, and 2.5 s in consecutive trials at a penetration rate of about 11 to 48 mm·s^−1^. In the case of the along-the-grain orientation, 60 mm long beech samples were penetrated over 1.2 s and 160 mm long samples over 6.02 s at a penetration rate ranging from about 27 to 50 mm·s^−1^ (the rate decreased with sample length).

For the 210 W laser beam, the penetration time of an oak wood sample with a diameter of 100 mm across the grain was 4.9 s through heartwood and 4 s through sapwood. The penetration time for an oak sample with a length of 50 mm along the grain was 1.49 s for heartwood and 0.5 s for sapwood, while for a 210 mm long sample, the corresponding values were 31 and 20 s, respectively. Birch wood samples with a diameter of 100 mm were penetrated across the grain with a 210 W laser beam over 5.12 and 2.58 s (at a rate from 19.4 to 38.8 mm·s^−1^). The penetration rate along the grain ranged from 16.34 mm·s^−1^ for a 260 mm sample to 29.6 mm·s^−1^ for a 45 mm sample. The highest penetration rate (4.92 mm·s^−1^) was found for a 160 mm birch wood sample. These initial trials were the motivation for further research, and a patent application for laser shears was filed by the authors (a patent was granted in 2019).

Unfortunately, the power of the laser used in the initial trials was too low to enable satisfactory cutting times. Therefore, further studies involved a CO_2_ laser of sufficient power, which was available to the authors. The CO_2_ laser was deemed suitable because the wavelength of the emitted beam was appropriate for wood, and similar lasers were already widely used in the woodworking industry.

Laser cutting experiments were conducted at the Center for Laser Technologies of Metals, Kielce University of Technology. SEM images of cut surfaces were analyzed at the Department of Biosystems Engineering, Warsaw University of Life Sciences. The device used in the experiments was a pulsed wave CO_2_ laser (Trumpf) emitting a maximum output power of 6 kW at 10,600 nm. The laser was equipped with a zinc selenide lens with a focal length of 5” (127 mm), with the beam focused on a spot with a diameter of 0.34 mm. During the trials, the distance between the laser head and the wood sample was 2 mm.

Experiments were conducted at three laser beam power levels: 2000, 1000, and 500 W at a frequency of 20 kHz. Three cutting rates were used: 1000, 500, and 300 mm∙min^−1^. The structure of the cut surfaces was examined using an FEI Quanta 200 ESEM scanning electron microscope at 500× and 1000× magnifications. The samples were appropriately prepared prior to microscopic examination.

The study involved four tree species: Scots pine (*Pinus sylvestris* L.), pedunculate oak (*Quercus robur* L.), black poplar (*Populus nigra* L.), and silver birch (*Betula pendula* Roth). The wood samples were obtained from a private forest in the municipality of Pniewy, Grójec County (GPS 51°52′49″ N, 20°43′39″ E), from an approximately 25-year-old stand that developed spontaneously on a former arable land, on what could be classified as a fresh mixed coniferous site.

Laser cutting trials involved segments of branches with bark freshly collected in late February and early March, with a diameter of 25, 45, and 65 mm. During the research, samples of wet (freshly cut) wood were cut and taken from growing trees about 2 h earlier.

The mean absolute moisture content was 163% for Scots pine, 69% for pedunculate oak, 109% for black poplar, and 102% for silver birch. The moisture content of the samples was determined by weight loss on drying. The initial and final weight was measured using a WPS210S laboratory balance (Radwag, Radom, Poland) with an accuracy of 0.001 g. The samples were dried in a Heraeus UT 6120 circulating air oven at a constant temperature of 105 ± 2 °C over 24 h (Kendro Laboratory Products GmbH, Hanau, Germany).

For each setting (*i*), the energy per 1 mm of cut surface was calculated as *E_ji_* (J∙mm^−1^).
(1)$$E_{j}=\frac{P}{v},$$
where

*P*—laser beam power (W),

*v*—beam travel speed (mm∙s^−1^).

The following parameters were applied in the process of cutting the studied wood samples of each tree species:

P1—power of 1000 W at a beam travel speed of 1000 mm∙min^−1^,

*E_j_* = 60 J∙mm^−1^;

P3—power of 500 W at a beam travel speed of 500 mm∙min^−1^,

*E_j_* = 60 J∙mm^−1^;

P7—power of 1000 W at a beam travel speed of 300 mm∙min^−1^,

*E_j_* = 200 J∙mm^−1^;

P8—power of 2000 W at a beam travel speed of 300 mm∙min^−1^,

*E_j_* = 400 J∙mm^−1^.

Each cut sample was photographed, and the images were analyzed using MultiScanBase v.18.03 software to measure the section area and circumference. Based on these data, the radius of each section was calculated on the assumption that the shape of the section was close to circular. The results naturally differed due to deviations of the actual section shape from the circles representing it. Thus, to decrease the approximation error, means were calculated from the two aforementioned radii, with the resulting value being termed the adjusted radius (*r*).

In order to determine the energy of laser radiation per 1 mm^2^ of the cut surface (*E_p_*), side length was calculated for a square with an area equal to the area of a circle with the radius *r* (*l_sq_*).
(2)$$E_{p}=\frac{E_{j}}{l_{sq}},$$

## 3. Results

Table 2 shows the adjusted radii (*r*), equivalent square side lengths (*l_sq_*), and mean laser radiation energy per 1 mm^2^ of cut surface (*E_p_*) for wood samples from four tree species and four laser-cutting settings.

It was found that the degree of cut surface charring increased with increasing power and decreasing beam travel speed (Figure 3). As can be seen, at a power of 1000 W and a beam travel speed of 1000 mm∙min^−1^ (P1), the sample section structure was not substantially altered. On the other hand, a high degree of surface charring occurred at a greater power and reduced beam travel speed, that is, 2000 W and 300 mm∙min^−1^, respectively (P8). The laser settings used for samples P1 and P3 were sufficient for cutting wood samples with diameters ranging from 16 to 70 mm (corresponding to the diameters of branches cut in orchards and during conifer pruning). Given the obtained cutting effectiveness, subsequent trials could be considered superfluous. However, if the charring of the cut surface is assumed to be an equally important goal of the laser cutting of fresh wood, then only the settings for trial P8 can be deemed effective. In trials P1, P7, and P8, the degree of charring was clearly correlated with the delivered radiation energy per unit of cut area (*E_p_*) for all the tree species.

However, a comparison of trials P1 and P3 produces puzzling results: even though the overall energy delivered to the cut surface was the same and the amount of energy per unit of area was smaller in P3 (given its larger diameter), P3 revealed a noticeably higher degree of charring. This suggests that the degree of charring may depend on the duration of workpiece exposure rather than laser beam power.

A microscopic image of the saw-cut surface of the black poplar at a magnification of 500× (Figure 4) shows numerous disrupted and frayed cells. It can be seen that the extent of damage differs between the earlywood and latewood areas, with greater damage in the former. Similar observations were made for the other tree species studied.

Figure 5 shows a SEM image of a black poplar sample cut with shears. While the extent of damage is smaller than in the case of saw cutting, the wood structure is still cracked and crushed in numerous places. Thus, following both saw and shear cutting, the wood tissue is exposed to an increased risk of pathogen invasion.

Figure 6 shows SEM images of Scots pine samples cut across and along the grain. The cross section of the pine wood sample cut with a 1000 W laser beam at a travel speed of 1000 mm∙min^−1^ reveals well-defined cells with regular walls. In turn, the image of the sample cut at a lower power setting and slower beam travel speed (500 W and 500 mm∙min^−1^, respectively) shows less well-defined cells with their walls disintegrating and fusing with adjacent cells. These results indicate that latewood cells are more affected than earlywood cells. A decrease in cutting rate to 300 mm∙min^−1^ at an increased power output (2000 W) led to substantial destruction of the cellular structure, with some cells exhibiting altered shapes and charred walls.

The surface structure of the longitudinal section of the sample cut with a 1000 W laser beam at a travel speed of 1000 mm∙min^−1^ is clear and smooth with well-defined cell walls and distinct cell shapes (Figure 6). A reduction in laser power and travel speed to 500 W and 500 mm∙min^−1^, respectively, led to deformed, blurred cells. In both cases, a layer of compacted cells was produced on the cut surface, which could provide a good barrier to pathogens. Laser cutting at a beam power of 2000 W and travel speed of 300 mm∙min^−1^ along the grain substantially damaged and charred the outer cell layer to a depth of approximately 50 µm (individual charred cells cannot be distinguished). It should be noted that the charred layer may also favorably affect the dynamics of pathogen invasion. The exact effects of the layers produced on the cut surface in terms of preventing pathogen infection require further study.

Interestingly, despite the fact that laser cuts made with a 1000 W beam at 1000 mm∙min^−1^ and with a 500 W beam at 500 mm∙min^−1^ delivered a similar overall amount of energy to the cut surface, wood cells were more altered in the latter case. This suggests that beam travel speed may have a greater effect on cellular structure than laser power.

Figure 7 shows the surface structure of laser-cut oak wood for the studied cases. Cutting at a high beam travel speed (1000 mm∙min^−1^) and power (1000 W) led to well-defined bisected oval cells with visible lumina (P1). In turn, the second image (P3) shows earlywood in an annual ring (large cells) and latewood fragments in successive rings, above and below them, cut with a laser beam at half the power (500 W) and beam travel speed (500 mm∙min^−1^). A comparison of latewood from the first and second images reveals that the structure of the latter partially disintegrated as a result of charring, with the cell walls being less well-defined than that in trial P1 involving a higher beam power and beam travel speed.

It was found that a decrease in beam travel speed from 500 to 300 mm∙min^−1^ in conjunction with a fourfold increase in beam power considerably damaged wood cells. The wood structure in P8 in Figure 7 is frayed, and one can hardly discern cell shapes, especially in the latewood (which is harder). This layer resembles a charred spongy structure.

P1 in Figure 7, presenting the longitudinal structure of a sample cut with a 1000 W laser beam at a travel speed of 1000 mm∙min^−1^, reveals a smooth cut surface with a compacted layer of cells up to the depth reached by the laser gasses. In turn, the sample cut with a beam power of 500 W at 500 mm∙min^−1^ (P3 in Figure 7) clearly shows a frayed layer of charred cells. Similarly, as in the case of pine wood, laser cutting at 2000 W and 300 mm∙min^−1^ led to considerable charring of the surface (the charred layer is approximately 50 µm thick with intact cells underneath).

Figure 8 compares the cellular structures of birch wood cut along the grain with a laser power of 1000 W at a travel speed of 1000 mm∙min^−1^ and with a power of 2000 W at a travel speed of 300 mm∙min^−1^.

Similarly, as in the other cases, cellular structure is clearly visible at a lower laser power, with a compacted cell layer on the surface. At a higher beam power and lower beam travel speed, the cut surfaces were charred.

Figure 9 presents a comparison of the cellular structures of poplar wood cut with a laser power of 1000 W at a travel speed of 1000 mm∙min^-1^ and with a laser power of 500 at a travel speed of 500 mm∙min^−1^.

Laser cutting with a higher beam power and travel speed across the grain led to a well-defined cell structure with a thin layer of compacted cells on the cut surface. At a lower beam power and slower travel speed, the wood cells are not as clearly visible as their structure is altered to a greater extent. Thus, the experimental results show that for both cut orientations (across and along the grain), cells are more prone to disintegration and charring at lower laser travel speeds.

The wide application of CO_2_ lasers in the furniture and packaging industry involves dried wood, having dissimilar physical and chemical properties to fresh, living wood. The motivation of this study was the possible reduction of the susceptibility of living wood to pathogens. This is now under examination, and the preliminary results look promising.

We are currently conducting research comparing the penetration of pathogens through wood surfaces cut by laser with that cut by saws or pruning shears. The preliminary results look promising. We will present the results of these studies in the next publication. In the future, we plan to conduct extended research using a fiber laser.

## 4. Conclusions

The presented preliminary results show that a CO_2_ laser may be successfully applied to cut fresh wood. However, further efforts should focus on fiber-optic and semiconductor lasers since the design of CO_2_ lasers makes them difficult to deploy in the form of portable shears.

The higher the power of the laser beam at a given travel speed, the greater the destruction of the cut surface (charring). However, when comparing two settings with the same overall amount of energy delivered to the sample, greater alteration of cellular structure occurs for the lower laser power delivered at a slower travel speed.

The surface of cuts made with a saw or shears reveals extensive cell damage and numerous cracks that enable pathogen invasion of the affected plants; furthermore, cell damage differs between latewood and earlywood areas. Those effects were not observed in the laser cutting trials.

## Figures and Tables

**Figure 1 materials-13-05222-f001:**
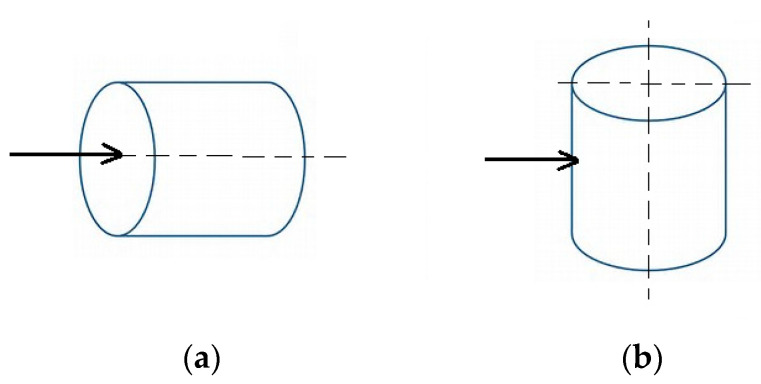
Diagram illustrating laser beam penetration through wood samples: (**a**) along the grain, (**b**) across the grain.

**Figure 2 materials-13-05222-f002:**
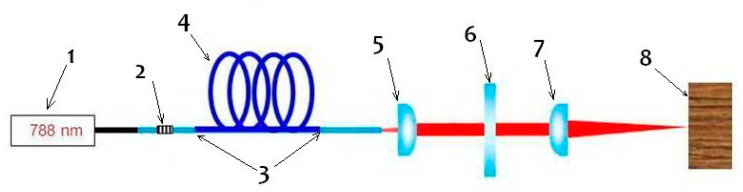
Diagram of the laser setup used in the preliminary study: (1) semiconductor pumping laser with a wavelength of 788 nm, (2) fiber Bragg grating, (3) fiber splice connection, (4) Tm-doped fiber, (5) collimator, (6) filter (HR 788 nm/AR 2 μm), (7) converging lens, and (8) wooden workpiece.

**Figure 3 materials-13-05222-f003:**
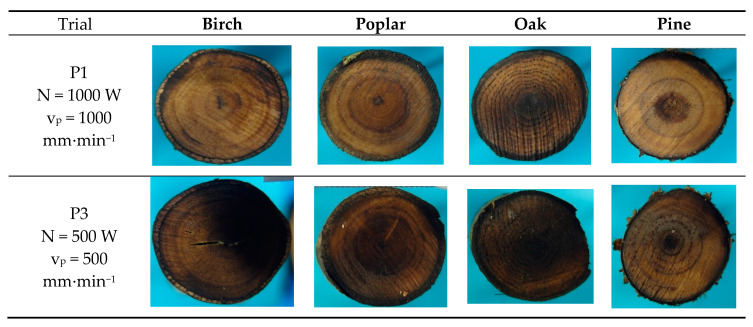
Cut surfaces at different laser beam powers and travel speeds.

**Figure 4 materials-13-05222-f004:**
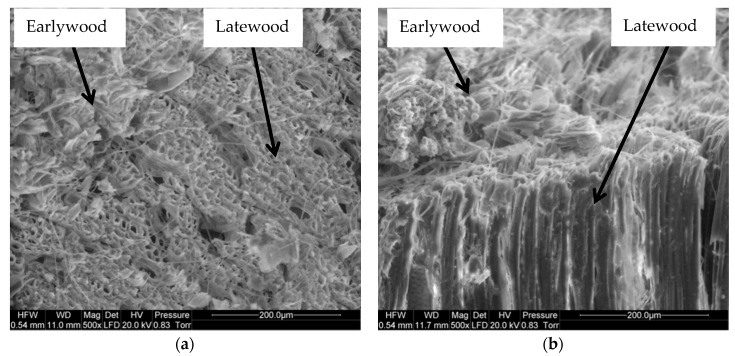
SEM image of a saw-cut surface of black poplar: (**a**) transverse section, (**b**) longitudinal section.

**Figure 5 materials-13-05222-f005:**
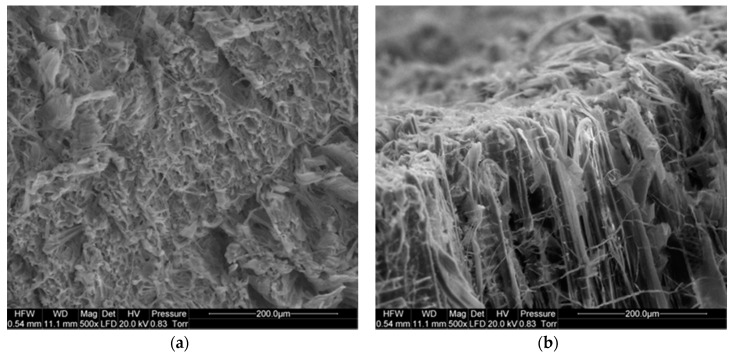
SEM image of a shear-cut surface of black poplar: (**a**) transverse section, (**b**) longitudinal section.

**Figure 6 materials-13-05222-f006:**
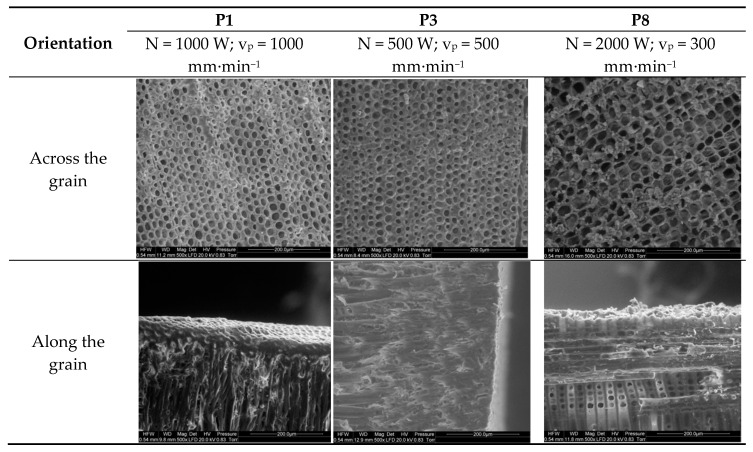
Surface structure of Scots pine samples laser-cut across and along the grain (trials P1, P3, and P8).

**Figure 7 materials-13-05222-f007:**
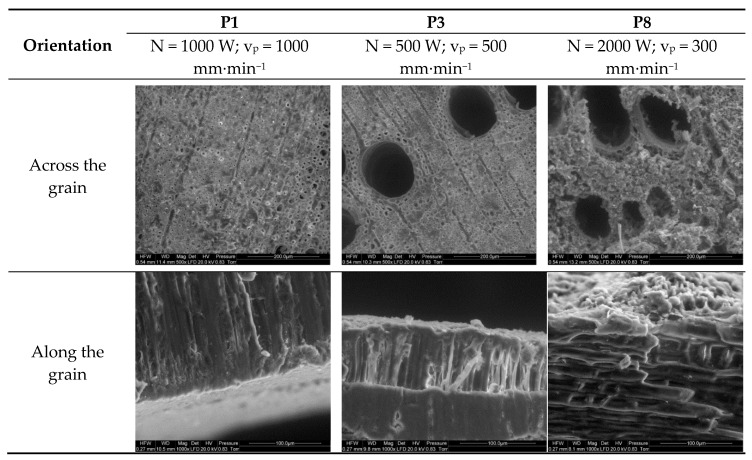
Surface structures of oak wood cut with a laser beam across and along the grain (trials P1, P3, and P8).

**Figure 8 materials-13-05222-f008:**
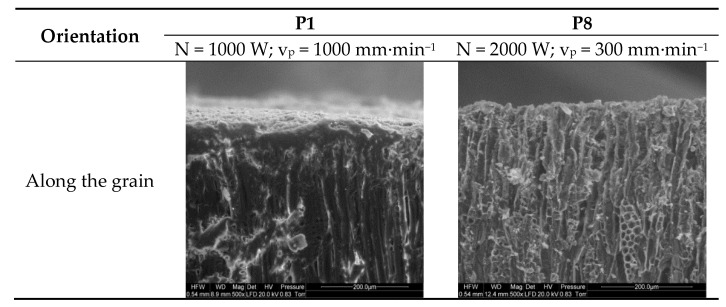
Comparison of birch samples laser-cut along the grain in trials P1 and P8.

**Figure 9 materials-13-05222-f009:**
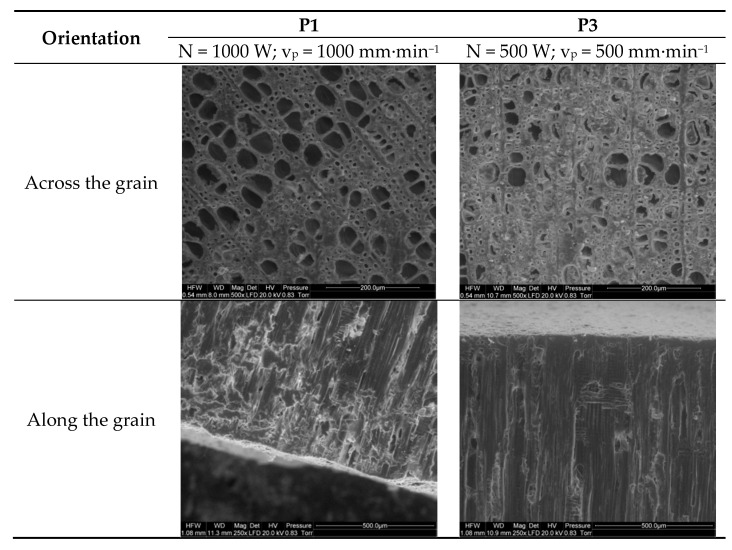
Cellular structure of poplar wood after laser cutting in trials P1 and P3.

**Table 1 materials-13-05222-t001:** The results of the measurements of the impact of the fiber laser beam on wood.

No.	Type	Direction of Breakthrough	Sample Diameter (mm)	Length Samples (mm)	Beam Power Laser (W)	Focal Length (mm)	Time Punctures (s)	Piercing Speed (mm s^−1^)	Focusing Lens
1	Pine	AC	120	-	150	-	The laser did not pierce	-	No
2	Pine	AC	120	-	150	500	129.0	0.93	Yes
3	Pine	AC	120	-	150	500	243.0	0.49	Yes
4	Pine	AC	120	-	210	230	3.0 5.0	40 24	Yes
5	Pine	WW	-	120	210	230	1.0 1.0	120 120	Yes
6	Beech	AC	120	-	210	230	11.0 4.0 2.5	10.90 30 48	Yes
7	Beech	AL	-	160	210	230	6.02	26.58	Yes
8	Beech	AL	-	60	210	230	1.20	50	Yes
9	Oak	AC	100	-	210	230	4.00 W 4.90 H	25 20.41	Yes
10	Oak	AL	-	50	210	230	1.49 H 0.50 W	33.56 100	Yes
11	Oak	AL	-	140	210	230	4.79 H 5.16 W	29.23 27.13	Yes
12	Oak	AL	-	210	210	230	31.00 H 20.06 W	6.77 10.47	Yes
13	Birch	AC	100	-	210	230	5.12 2.58	19.53 38.76	Yes
14	Birch	AL	-	45	210	230	1.52	29.61	Yes
15	Birch	AL	-	160	210	230	3.91	40.92	Yes
16	Birch	AL	-	260	210	230	15.91	16.34	Yes

AC—across the fibers, AL—along the fibers, H—heartwood, W—whiteness of the wood.

**Table 2 materials-13-05222-t002:** Adjusted radii (*r*), equivalent square side lengths (*l_sq_*), and mean laser beam energies per 1 mm^2^ of cut area (*E_p_*) for four tree species and four laser settings.

Trial	Parameter	Birch	Poplar	Oak	Pine
P1	Adjusted radius *r* (mm)	8.8	10.5	10.9	9.6
N = 1000 W	Square side length *l_sq_* (mm)	15.6	18.5	19.2	17.1
v_p_ = 1000 mm·min^−1^	Laser beam energy *E_p_* (J∙mm^−2^)	3.85	3.24	3.12	3.51
P3	Adjusted radius *r* (mm)	10.5	12.4	11.7	12.7
N = 500 W	Square side length *l_sq_* (mm)	18.6	21.9	20.7	22.6
v_p_ = 500 mm·min^−1^	Laser beam energy *E_p_* (J∙mm^−2^)	3.22	2.74	2.90	2.65
P7	Adjusted radius *r* (mm)	24.0	19.3	18.4	20.7
N = 1000 W	Square side length *l_sq_* (mm)	42.6	34.3	32.7	36.7
v_p_ = 300 mm·min^−1^	Laser beam energy *E_p_* (J∙mm^−2^)	4.69	5.83	6.12	5.45
P8	Adjusted radius *r* (mm)	35.1	32.9	35.9	31.2
N = 2000 W	Square side length *l_sq_* (mm)	62.1	58.2	63.6	55.2
v_p_ = 300 mm·min^−1^	Laser beam energy *E_p_* (J∙mm^−2^)	6.44	6.87	6.29	7.24

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
