# Peer review of "Infrared Laser Application to Wood Cutting"

_materials, 2020, doi:10.3390/ma13225222_

Round 1

Reviewer 1 Report

This manuscript is on the investigation of IR laser cutting of wood, which has very good practical applications in our life. Wood cutting, especially for fresh wood with water, is challenging. Right laser sources and laser processing parameters are critical for high quality cutting. The authors did extensive experiment and results are interesting and reliable. The manuscript can be accepted for publication after the following revisions:

1. Figures 1 and 2 may be combined into one figure on wood optical property.

2. Figure 3 and 4 may be combined into one figure on experimental setup.

3. To add “,” or “.” After one equation.

4. There are sound generated during the laser processing. The authors can set up a real-time monitoring system to control the laser wood cutting. The following paper is a good reference: Papanikolaou A, Tserevelakis G J, Melessanaki K, Fotakis C, Zacharakis G et al. Development of a hybrid photoacoustic and optical monitoring system for the study of laser ablation processes upon the removal of encrustation from stonework. Opto-Electron Adv 3, 190037 (2020). 

5. It would be much better if the authors can provide mechanical and chemical properties of the wood samples after the laser processing.

6. To enhance laser cutting speed is critical to high productivity, optimal laser wavelength is important.  3um MIR laser would be the better light source for wood cutting. The authors can comment this issue by referring the following paper: Zhang X J, Li W W, Li J, Xu H Y, Cai Z P et al. Mid-infrared all-fiber gain-switched pulsed laser at 3 μm. Opto-Electron Adv 3, 190032 (2020). 

Reviewer 2 Report

This manuscript communicates some interesting aspect of wood cutting using infrared laser pulses. It is very well written with relatively good clarity and technical soundness. The authors start from fiber laser, and then step-by-step experiments all way through CO2 laser to the final conclusions, with some good quality SEM images as support. 

However, I do have a strong feeling that in terms of novelty and impact the manuscript in its current form is not up to par. The introduction was written in a way that one would expect a scientific discussion in the field of fiber laser application in wood cutting/trimming/pruning, which, would have been a good shot for publication. However, this is just an impression but not the case. The majority work is on CO2 laser wood cutting, which is, as the authors put it themselves, a technology having been around for 50 years. I haven’t done an exhaustive research on the state of the art of CO2 laser cutting woods. Nonetheless, I am not convinced that the study of power and speed is a breakthrough discovery, unless the authors could defend themselves in their revision.  

The potential of reduced pathogen invasion is a novel concept. However, it remains as a perspective, rather than a fact. Figures 8 to 11 are very impressive, but they do carry new information. A dedicated study on this issue would definitely worth a publication credit.

The manuscript is a bit disorganized and some experimental results are difficult to understand. Here come a few detailed questions:

  • How is the laser cavity in Fig 4 achieved with only 1 FBG?
  • Results presented in methodology section are very disorganized. A table would greatly improve their understanding. In addition, the relation of these results with the finalones should be discussed in higher detail.
  • Check Fig.4 in line 128. It looks like there is a typo and a missing figure.
  • One of the motivations for this study is to prove that CO2 can be successfully employed in fresh wood cutting. In such case, why were samples dried? wouldn't that change the results?
  • P1-P8 are explained in page 5, below eq. 1. However, it would be handy to repeat such parameters in table 1 or fig. 5 for a clearer result interpretation.
  • A combustion threshold velocity at different powers should be calculated.
  • Minor English and format issues here and there

Given the facts listed above, I am hesitating to give a rejection or a substantial major revision.  If a revision is the case, the authors have to shake up the form of the manuscript. The part about fiber laser has to be considerably shortened, as it is not the core of the report. The results on CO2 laser has to be much more convincing, more rigorous literature review and results discussion are needed in this case. Possible connection to pathogen invasion test would be appreciated.

Reviewer 3 Report

Dear authors,

the manuscript deals with the influence of laser power and laser beam speed of an infrared laser on cut surfaces of different wood samples. The manuscript is written very well. So, I require to perform a minor revision of the manuscript and the publication of the manuscript after its improvements based on my few requirements.

- In Abstract, I recommend to mention which type of laser was used during the experiment.

- Line 36 - Authors mentioned so called „MDF cutting“. Could authors explain the abbreviation "MDF"?

Round 2

Reviewer 2 Report

Thanks for the authors revision. The quality has been improved noticeably. I have a few minor comments for the authors to make a further effort:

-  The following content from the author’s response could be re-written and fitted into the manuscript before the conclusion section to serve as a perspective for the readers to expect something. The manuscript, albeit improvement, looks still a bit like an experiment report, rather than a scientific paper.

The wide application of CO2 lasers in furniture and packaging industry involves dried wood, having dissimilar physical and chemical properties to the fresh, living wood. The motivation of this study was possible reduction of susceptibility of the living wood on pathogens. This is now under examination, the preliminary results look promising.

We are currently conducting research comparing the penetration of pathogens through wood surfaces cut by laser to those cut by saws or pruning shears. The preliminary results look promising. We will present the results of these studies in the next publication. In the future, we plan to conduct extended research using a fiber laser.

- The arrows and “earlywood” “latewood” annotations can be embedded into the SEM micrographs. It would look nice and neat.

- The SEM images in Figures 8 and 9 are not well aligned, at least in the pdf version that I am reading.

- The 2 in “CO2 laser” in Abstract section should be a subscript instead of normal sized letter.

Author Response

Thank you for the new reviews.

Changes resulting from the reviewer's comments were introduced to the manuscript and marked with a yellow background. Corrected figures are marked with appropriate comments. The whole paper has been re-edited according to the publisher's recommendations.